# Endemic Freshwater Fish Range Shifts Related to Global Climate Changes: A Long-Term Study Provides Some Observational Evidence for the Mediterranean Area

**Antonella Carosi** [1,*] **, Rosalba Padula** [2]**, Lucia Ghetti** [3] **and Massimo Lorenzoni** [1]

[1]   Department of Chemistry, Biology and Biotechnologies, University of Perugia, via Elce di Sotto,
      06100 Perugia, Italy; Massimo.lorenzoni@unipg.it
[2]   Center "Climate Change and Biodiversity in Lakes and Wetlands" of Arpa Umbria, 06100 Perugia, Italy;
      r.padula@arpa.umbria.it
[3]   Forest, Economics and Mountain Territory Service, Umbria Region, 06100 Perugia, Italy;
      lghetti@regione.umbria.it
*   Correspondence: antonella.carosi@unipg.it; Tel.: +39-075-5855716

**Abstract:** Climate changes will lead to a worsening of the ecological conditions, in terms of hydrological instability and rising water temperatures, of the Mediterranean rivers. Freshwater fishes inhabiting this area can be threatened in the near future by accelerating drought and decreased ecological connectivity. The main aim of the research was to analyze changes in the distribution of the endemic freshwater fishes *Padogobius nigricans*, *Squalius lucumonis* and *Telestes muticellus* in the Tiber River basin (Italy), within a proven period of climate warming, in terms of increasing water temperature and droughts. A multivariate analysis was conducted using fish and environmental data collected in 117 sites over the years 1990–2017. For the three species, population abundance, age structure and body condition were analyzed. Detectability, occupancy, local extinction and colonization processes were also examined. We showed that *S. lucumonis* and *T. muticellus* have shifted their distributions upstream, likely in order to reach their thermal optimum. *Padogobius nigricans* did not move upstream significantly, since the species is characterized by limited vagility and thus a low dispersal capability in a context of high river fragmentation. In the study area, elevation and river barriers seem to play a key role in extirpation and colonization processes; for *S. lucumonis* and *T. muticellus* the extinction probability decreased with increasing altitude, while for *P. nigricans* the colonization probability decreased with an increasing degree of river fragmentation. These results highlight how species-specific dispersal ability can lead to varying adaptability to climate change.

**Keywords:** biodiversity conservation; endemic fish species; climate change; fish range shifts; river connectivity

---

## 1. Introduction

The Mediterranean area is considered a biodiversity hotspot due to the presence of a large number of endemic species [1], which includes many freshwater fishes [2]. In particular, inland waters are characterized by high levels of fish biodiversity [3]. Considering the limited dispersal ability of primary fish species (see Myers [4]), freshwater ecosystems are particularly vulnerable to many anthropogenic stressors, because they represent more isolated and fragmented environments than terrestrial ecosystems [5]. Habitat alteration, invasive species, water pollution and abstraction, and over fishing are among the main threats [3] leading to 56% of endemic freshwater fish in the Mediterranean

basin being threatened [2]. The negative effects of these multiple stressors can be enhanced by global climate change [6–8]. Climate changes will in effect lead to a worsening of ecological conditions, in terms of hydrological instability and decreased connectivity, of the Mediterranean watercourses [9]. As a result of climate change, increasing drought and the progressive deterioration of water quality in aquatic environments of this area can be predicted in the near future, with a consequent decrease in freshwater biodiversity [10].

Climate change can produce direct effects on fish at individual, population, species, and community levels [11]. With regards to the species level, different fish species do not react in the same way [12,13]; some species are much more susceptible to the impacts of climate change than others, due to different biological, ecological features and various life-history strategies [14]. In particular, the distribution of the most vulnerable species could be largely influenced by climate-induced effects such as rising temperatures and water availability [7,15,16]. In fact, as a short-term response to climate change, species can look for the most appropriate environmental conditions and try to follow them moving in space [17,18]. In watercourses, these displacements mainly occur in a longitudinal direction, given the presence of an upstream–downstream gradient that strongly affects the structure of fish assemblages [19,20]. As a result of increases in water temperature, caused by decreased rainfall and increases in water abstraction and evapotranspiration, some studies revealed a tendency for fish species to move upstream, towards a thermal optimum [7,21,22]. However, it should be considered that species' movements upstream can be hindered by insurmountable barriers or by unsuitable environmental conditions, and that it may be necessary for moving species to establish new biotic interactions within newly colonized habitats [23,24]. In this case, the species that will not be able to adapt to new conditions will be at risk of local extinction [8].

The water bodies of Central and Southern Italy are particularly threatened by climate change. Flow rates of the Apennine chain watercourses are closely linked to atmospheric precipitations. Climate models predict an exacerbation of their torrential characteristics, with an intensification of extreme meteorological phenomena and reduced flow rates in the summer season [9]. This situation is further aggravated by increases in water withdrawals for irrigation purposes [25,26]. The Tiber River basin (Central Italy) is characterized by the presence of a large number of endemic fish species with limited areal distribution which are, therefore, highly vulnerable to extinction [27,28]. In fact, the loss of flow occurring in the Tiber River basin during periods of drought can lead to the isolation of endemic species with limited thermal tolerance. Three species in particular are subject to the potential effects of climate change in this area: the Etruscan chub *Squalius lucumonis* (Bianco 1983), the Italian riffle dace *Telestes muticellus* (Bonaparte, 1837), and the Arno goby *Padogobius nigricans* (Canestrini, 1867). *Squalius lucumonis* is an endemic species with a distribution limited to three river basins in Central Italy [29,30]; the species is listed as endangered on the International Union for Conservation of Nature (IUCN) red list of threatened species [31], and the progressive reduction of the species range is due to habitat alterations and water withdrawals [30]. *Telestes muticellus* is endemic to Northern and Central Italy; it prefers moderately cool waters in upper stream reaches [32]. *Padogobius nigricans* is a species endemic to Central Italy. It is listed as vulnerable in the IUCN red list of threatened species [31] because it is threatened by habitat alterations and competition with the Padanian goby *Padogobius bonelli* (Bonaparte, 1846) [33,34], a species endemic to Northern Italy introduced in the Tiber river basin in 1993 [35]. These three species inhabit small watercourses in the upper part of the Tiber basin, which represent refuge areas in comparison with downstream reaches and lowland rivers, which are compromised by the combined action of water pollution and presence of many exotic species [23]. However, these refuge areas have marked torrential features and are subject to drought periods during the dry seasons. Currently there is scarce information in the literature on the thermal niche of the three species; however many studies showed their reophilic characteristics and their location in the middle-upper section of the rivers, where current speed and dissolved oxygen are quite high and water is mildly cool [27–29]. Among all the fish species occurring in the study area with limited thermal tolerance, *P. nigricans*, *T. muticellus* and *S. lucumonis* seemed to be particularly suitable for evaluating

the fish distribution shifts in freshwater ecosystems, since they are not a target for sport fishing, and they are not subject to stocking activities.

At present, there are not many studies concerning freshwater fish distribution range shifts linked to climate change, especially based on extended series of empirical data. Many evaluations regarding this topic are based only on the application of predictive models. In the present study, long-term environmental and fish observational data series were used to test the potential effects of climate changes on freshwater fish habitat preferences. In particular, the research aims were: (i) to highlight some environmental effects of climate change in watercourses of the upper section of the Tiber River basin; (ii) to analyze some changes of ecological preferences and the distribution range of *P. nigricans*, *T. muticellus* and *S. lucumonis* over time, probably related to climate warming.

## 2. Materials and Methods

### 2.1. Study Area

The Tiber River is the third-longest river in Italy and has the second-largest watershed, with a total surface of 12,692 km$^2$. The study area comprised 62 watercourses of the upper Tiber River basin (Figure 1) and encompassed a surface of 9413 km$^2$, equal to 55% of the total drainage area. The study area includes four sub-basins corresponding to the main tributaries of the Tiber River: Chiascio, Nera, Nestore and Paglia River basins. Our analyses utilized data collected during the census periods 1990–1997, 1998–2004, 2005–2011 and 2012–2017, in 117 sampling sites. All the sampling sites were sampled once for each census period, in autumn.

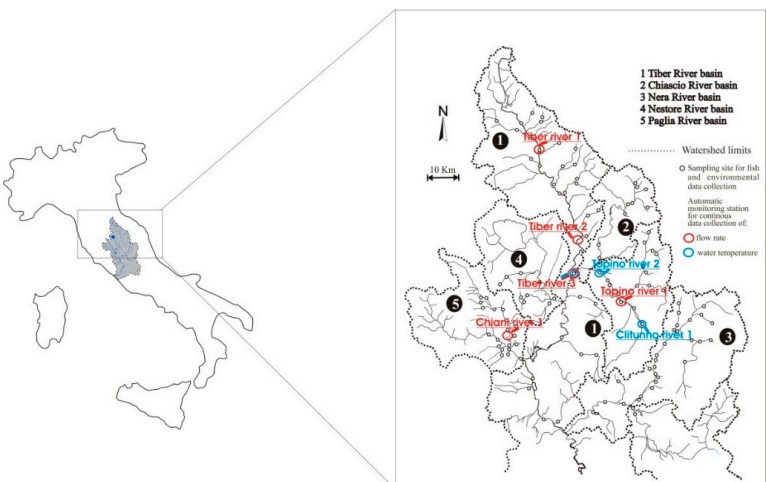

**Figure 1.** Study area and location of the sampling sites.

From a hydro-geological point of view the study area is mainly characterized by low permeability and marked flow rate oscillations; the Nera River basin represents an exception because, being composed of highly permeable carbonate rock complexes, it guarantees regular water conditions. The complete hydrographic network is characterized by the presence of numerous interruptions of river continuity, such as human-made weirs and dams. Agricultural land occupies about 50% of the study area. This aspect significantly affects the exploitation of surface water for irrigation purposes, especially during the summer season.

### 2.2. Data Collection

Relative to the census periods 1998–2004, 2005–2011 and 2012–2017, a census of the fish fauna by two-pass electrofishing was carried out at each sampling location, using the removal method [36,37]. Sampling was conducted during low flow periods using a continuous or pulsed current with power between 600 and 4000 W. Each river stretch of length between 50–100 m was sampled from downstream

to upstream direction twice consecutively, applying the same level of fishing effort [38]. All captured fishes were identified and counted in order to estimate the density (ind m$^{-2}$) and the standing crop (g m$^{-2}$) for each population. For all specimens, total length (TL) and weight (W) were measured and a scale sample was collected to determine age. Data collection from 1990 to 1997 comprised single-pass electrofishing data and therefore fish population density was not available, despite this is was important to include the data to have a longer time series. Therefore a comparable approach using species abundance uses a coding from zero to three based on the following categories: 0 = absent; 1 = rare; 2 = common; 3 = dominant.

Given that habitat features strongly influence the distribution of fish species and the composition of fish assemblages [39,40], 16 environmental parameters were used to characterize river stretches. Conductivity (μS cm$^{-1}$), pH (units), water temperature (°C) and dissolved oxygen (mg L$^{-1}$) were measured in the field at the same time of fish sampling, using electronic meters manufactured by YSI, Hanna Instruments and WTW GmbH. The hydrological parameters (flow rate m$^3$ s$^{-1}$ and current speed m s$^{-1}$) were measured at the cross-sectional area of each sampling reach, using an OTT MF-pro electromagnetic current meter manufactured by OTT Hydromet. Other chemical parameters of water (chlorides, sulphates, phosphates, ammonia) were subsequently measured in the laboratory according to the environmental protection agencies specifications [41–43]. Water samples were collected into 1 L polyethylene bottles at the same time of fish sampling from the water surface, as the depth values ranged from 0.07 to 199.00 cm (mean ± standard deviation (SD) = 0.35 ± 0.24). All samples were transported on ice to the laboratory. Watershed area (km$^{-2}$), distance from the source (km), and altitude (m a.s.l.) were derived from digital maps (geographic information system (GIS)) provided by the Forest, Economics and Mountain Territory Service of the Umbria Region. In order to evaluate water quality based on the presence of macro-invertebrates, the Extended Biotic Index (EBI) was used [44] (for more details, see [45]). To examine the opportunity for fish species to move upstream, the presence of dams and weirs located in the hydrographic network were detected and their geographical coordinates were derived using a GARMIN global positioning system (GPS) map 62 s with accuracy <10 m. The river fragmentation degree has been codified as the number of weirs with height >80 cm present downstream of each sampling site, up to the main receptor watercourse.

The Hydrographic Service of the Umbria Region provided continuous data of water temperature (°C) and flow rate (m$^3$ s$^{-1}$); both parameters were measured with automatic monitoring stations. Water temperature data was collected hourly from 2005 to 2016, at three sampling sites located on the Tiber, Topino and Clitunno Rivers, coded as Tiber River 3, Topino River 2 and Clitunno River 1, respectively. Flow rate data was collected hourly at five sampling sites: three sites were located on the Tiber River (coded as Tiber River 1, 2, 3), one site on the Topino River (coded as Topino River 1), and one site on the Chiani River (Chiani River 1) (Figure 1). The stations were chosen based on the representativeness of the entire hydrographic network and of the completeness of available data series. Flow rate calculations were performed using average monthly data.

*2.3. Body Condition and Size Structure Estimation*

In order to highlight changes in body condition over time, for *S. lucumonis* and *T. muticellus* relative weight (Wr) was assessed using the following equation: Wr = 100 (W/Ws) where body weight = W (g) and standard weight = Ws (same species in good physiological condition). The relative weight (Wr) is a condition index based on the comparison between the real weight of an individual, and the optimal weight of a specimen of the same species in good physiological condition (Ws). Wr values lower than 95 indicate poor body condition [46,47]. Wr estimation allows evaluation of the physiological status of a fish [48] and to highlight the occurrence of ecological changes over time [47]. In the present study the standard weight Ws was calculated using the following equations calculated with the empirical percentile (EmP) method [49,50].

*S. lucumonis*: $\log_{10}(Ws) = -7.750 + 5.750 \log_{10}TL - 0.660 (\log_{10}TL)^2$; TL application (cm) range 7–34,

*T. muticellus*: $\log_{10}(Ws) = -3.706 + 1.685 \log_{10}TL - 0.349 (\log_{10}TL)^2$; TL application (cm) range 6–17.

For *P. nigricans*, since the equation for calculating Ws is not available, body condition was estimated using the relative condition factor (Kn) [51] using the equation: $Kn = 100 W/(a\ TL^b)$, where a and b are the regression coefficients of the length–weight relationship (LWR) equation estimated from the total sample by the least-squares method [52], based on the logarithmic equation: $\log_{10}W(g) = a + b \log_{10}TL$ (cm). In order to avoid biases related to the accuracy of the weight measure, only fish of a total length larger than 4.0 cm were used to perform the analysis.

In order to provide a numeric estimation for deviations of the population structure from a balanced population, for *P. nigricans*, *S. lucumonis* and *T. muticellus* the proportional stock density (PSD) index [53] was calculated as follows:

PSD = 100 (number of fish ≥ minimum quality length/number of fish ≥ minimum stock length).

For *T. muticellus*, for minimum quality length and minimum stock length were used the values indicated by Pedicillo [54] for the populations of Central Italy. For *P. nigricans* and *S. lucumonis*, the minimum length thresholds for index calculations were established adopting the percentages suggested by Gabelhouse [53] and calculated on the basis of the largest specimen in the data set. The PSD values varied from 0 to 100 and the optimal range for a balanced population was: 35 ≤ PSD ≤ 65 [53,55].

### 2.4. Data Analysis

In order to evaluate temporal changes in water temperature and flow rate (dependent variables), linear regressions with years (independent) were performed. The significance of each relationship was tested using analysis of variance (ANOVA); the significance of the regression coefficient was tested using *t*-test analysis.

Since for *P. nigricans*, *T. muticellus* and *S. lucumonis* the population densities, the body condition and size structure indices were measured at the same sites across 3 periods (1998–2004, 2005–2011 and 2012–2017), in order to test their changes over time a one-way repeated-measures ANOVA was performed. Data were tested for sphericity using the Mauchly test, and the Greenhouse-Geisser and Huynh-Feldt adjustements were used. All the statistical analysis mentioned above were performed using Dell STATISTICA 13 software (Dell Inc, Aliso Viejo, CA, USA) for Windows.

In order to analyze the relationships amongst environmental and fish species data matrices for the four sampling periods (1990–1997, 1998–2004, 2005–2011 and 2012–2017), canonical correspondence analysis (CCA) [56] was performed. The CCA was processed with the CANOCO statistical package for Windows 4.5. Following Lepš and Šmilauer [57], since environmental and biological data were in the form of repeated measurements, the analysis was performed using time (i.e., census period) as a covariate. In order to avoid biases relate to the unimodality of the method, five sampling sites located less than a kilometer from a weir were excluded from the analysis. The environmental matrices included 16 variables (altitude, average current speed, chlorides, conductivity, dissolved oxygen, distance from the source, EBI, EBI quality class, flow rate, $NNH_3$, pH, $PPO_4$, river fragmentation degree (units), sulphates, water temperature, watershed area) and 112 observations (sampling sites). The fish matrices included 42 variables (fish species) and 112 observations (sampling sites). In order to make the data of the abundances detected in the four census periods comparable, for the years 1998–2004, 2005–2011 and 2012–2017, the species' abundances were coded using a scale ranging from zero to three based on the population density (0 ind $m^{-2}$ = absent; <0.05 = rare; 0.05 to 0.1 = common; >0.1 = dominant). All the variables (N) used in the analysis were transformed [$\log_{10}(N + 1)$] to normalize their distributions [58] and standardized to a mean of zero and standard deviation of 1. Statistical significance was tested using the Monte Carlo permutation test (999 permutations). For *S. lucumonis*, *P. nigricans* and *T. muticellus*, differences in average altitude values among all four sampling periods were tested using ANOVA, taking into consideration only the sites in which the species were present. ANOVA was performed using Dell STATISTICA 13 software for Windows.

Differences in water temperature preferences amongst sampling periods 1990–1997 and 2012–2017 were tested using *t*-test analysis. For the same species, with the aim to graphically represent their response curves in relation to altitude, and to highlight changes over time, a generalized linear model (GLM) was performed for each census period. GLM was performed with the CANOCO statistical package for Windows 4.5.

A multi-season model was used to estimate the occupancy, colonization, extinction and detection probabilities for *S. lucumonis*, *T. muticellus* and *P. nigricans* in the study area. The estimation was performed using presence-absence data for 117 survey sites across four periods (period 1: 1990–1997; period 2: 1998–2004; period 3: 2005–2011; period 4: 2012–2017). Elevation and degree of river fragmentation were used as site covariates, while water temperature was used as sample covariate. Akaike's information criterion (AIC) [59], was used to select the best candidate model. Following Burnham and Anderson [60], all models within 2 AIC units were considered. The analysis was processed using the program PRESENCE for Windows.

## 3. Results

### 3.1. Climate Changes

All three rivers showed significant, increasing trends in rising water temperatures over a 11 years period (Figure 2). Regression coefficients resulted highly statistically significant at the *t*-test analysis ($p < 0.001$). Water temperature increased from the period 2005–2010 to 2011–2016 by a mean of 0.4 °C for Topino ($y = 6.21 + 0.0002x$; $r = 0.05$; $F = 213.29$; $p = 0.0001$) and Tiber ($y = 4.83 + 0.001x$; $r = 0.05$, $F = 242.39$; $p = 0.0001$) and by 0.8 °C for Clitunno ($y = 10.47 + 0.0006x$; $r = 0.22$; $F = 3733.83$; $p = 0.0001$).

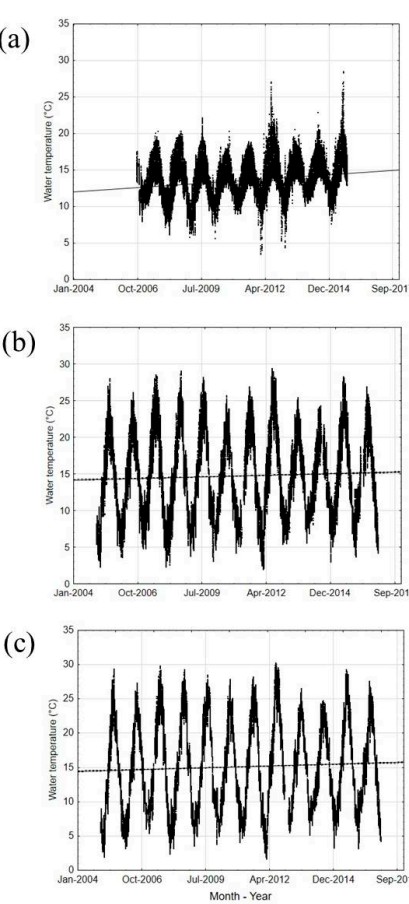

**Figure 2.** Trend over time of water temperature for 3 automatic monitoring stations: Clitunno River 1 (**a**), Topino River 2 (**b**), Tiber River 3 (**c**). Time periods: 2006–2015 for Clitunno River; 2006–2015 for Topino and Tiber Rivers.

The annual average flow rates showed a clear decreasing trend over time in the Chiani and Topino rivers than for the Tiber River. In both cases, the time series of data available (years 1992–2016) was shorter than the Tiber River (years 1920–2016) (Figure 3). All relationships resulted as statistically significant at the ANOVA analysis ($p < 0.05$); the regression coefficients resulted statistically significant at the *t*-test analysis ($p < 0.05$).

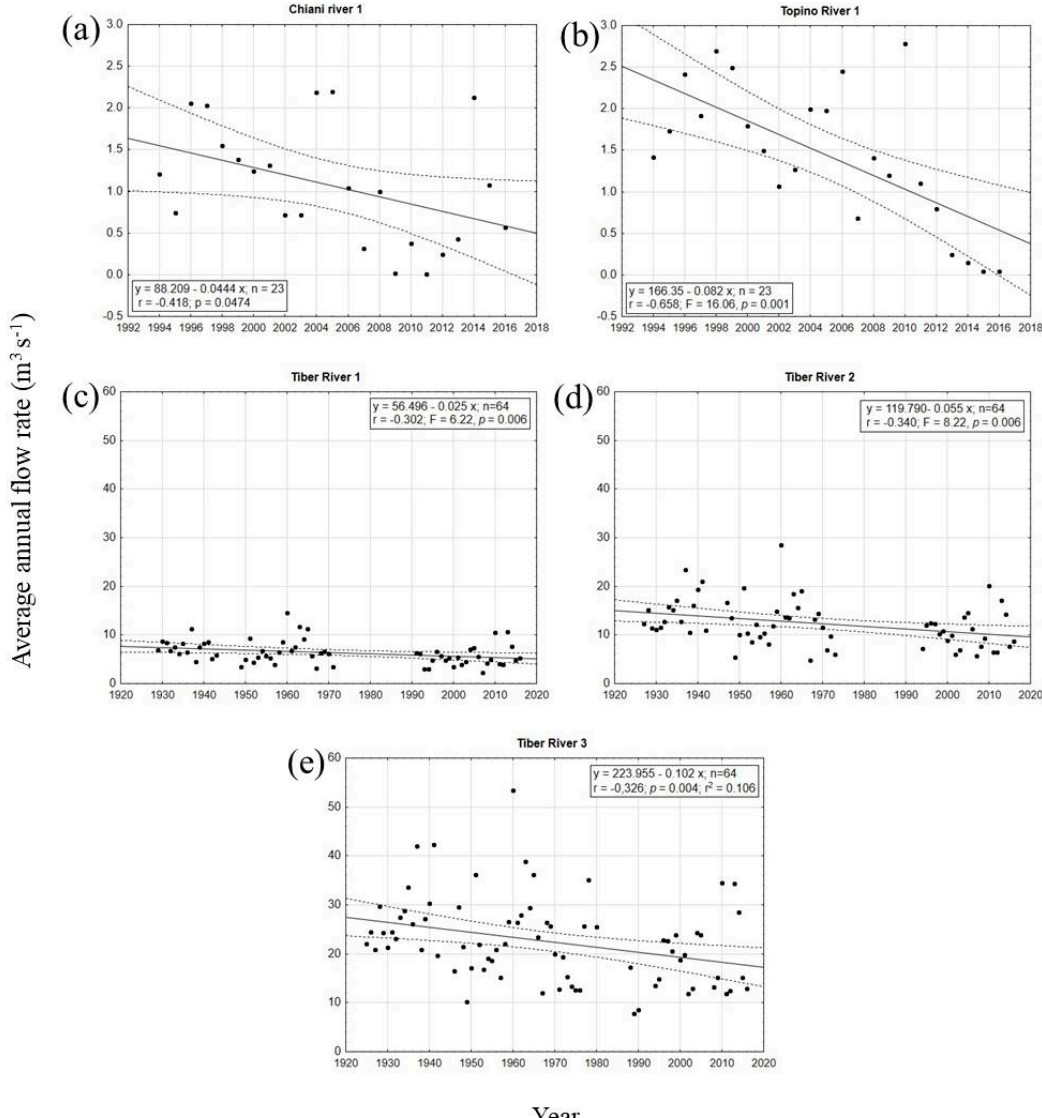

**Figure 3.** Trend over time of the annual average flow rate for 5 hydrometric sites: Chiani River 1, Topino River 1, Tiber River 1, 2, 3. Time periods: (**a**,**b**) 1992–2016; (**c**–**e**), 1920–2016. Solid lines represent the linear regressions. Dashed lines represent the regression bands with 0.95 confidence intervals.

### 3.2. Distribution Shift

The comparison between distribution maps related to the census periods 1990–1997 and 2012–2017 demonstrated the changes that occurred over time in the distribution of the three endemic species (Figure 4). *S. lucumonis* and *T. muticellus* appeared to extend their distribution range to some tributaries of the Tiber River in the northeast part of the basin, whilst the disappearance of *S. lucumonis* from the middle section of the Tiber River was detected. The number of sites all three fish species were increased from time period 1 (1990–1997) to time period 2 (2012–2017) by 4 sites for *S. lucumonis*, 10 sites for *T. muticellus* and 51 for *P. nigricans*. For *S. lucumonis* and *T. muticellus* the occupancy-elevation plots provided evidence for upper range limits expansion over time (Figure 5a,b). For *P. nigricans* the

plot suggested an overall range expansion of the species over time (Figure 5c). Over the four census periods (1990–1997, 1998–2004, 2005–2011 and 2012–2017) the comparison among the mean altitude values of sites in which the species were present, carried out by ANOVA, showed that *S. lucumonis* and *T. muticellus* distribution ranges significantly shifted to higher altitudes, resulting in a mean increase of 40.5 and 33.7 m, respectively (Figure 6a,b). In both cases, the ANOVA results were statistically significant (*S. lucumonis*: $F_{3, 252} = 1.96$, $p = 0.02$; *T. muticellus*: $F_{3, 366} = 2.32$, $p = 0.02$). By contrast, the distribution range of *P. nigricans* did not significantly move upstream ($F_{3, 176} = 0.32$, $p = 0.81$) (Figure 6c). To highlight the thermal conditions over time of the sites inhabited by the species, that could be related to their ability to shift, the mean water temperature values between the older census period (1991–1997) and most recent years (2012–2017) were compared using t-test analysis. The comparison resulted highly statistically significant for *P. nigricans* (1991–1997 mean value ± SD = 12.29 °C ± 5.23, range 2.0–20.0 °C; 2012–2017 mean value ± SD = 15.23 °C ± 3.93, range 6.0–23.7; $t = 3.01$, $p = 0.003$). For *S. lucumonis* (1991–1997 mean value ± SD = 14.95 °C ± 5.51, range 3.0–23.6 °C; 2012–2017 mean value ± SD = 15.01 °C ± 3.10, range 10.4–23.2; $t = 0.05$, $p = 0.956$) and *T. muticellus* (1991–1997 mean value ± SD = 13.53 °C ± 4.49, range 2.0–22.7 °C; 2012–2017 mean value ± SD = 13.87 °C ± 3.53, range 6.0–23.2; $t = 0.43$, $p = 0.672$) the *t*-test analysis did not show significant differences between the two periods.

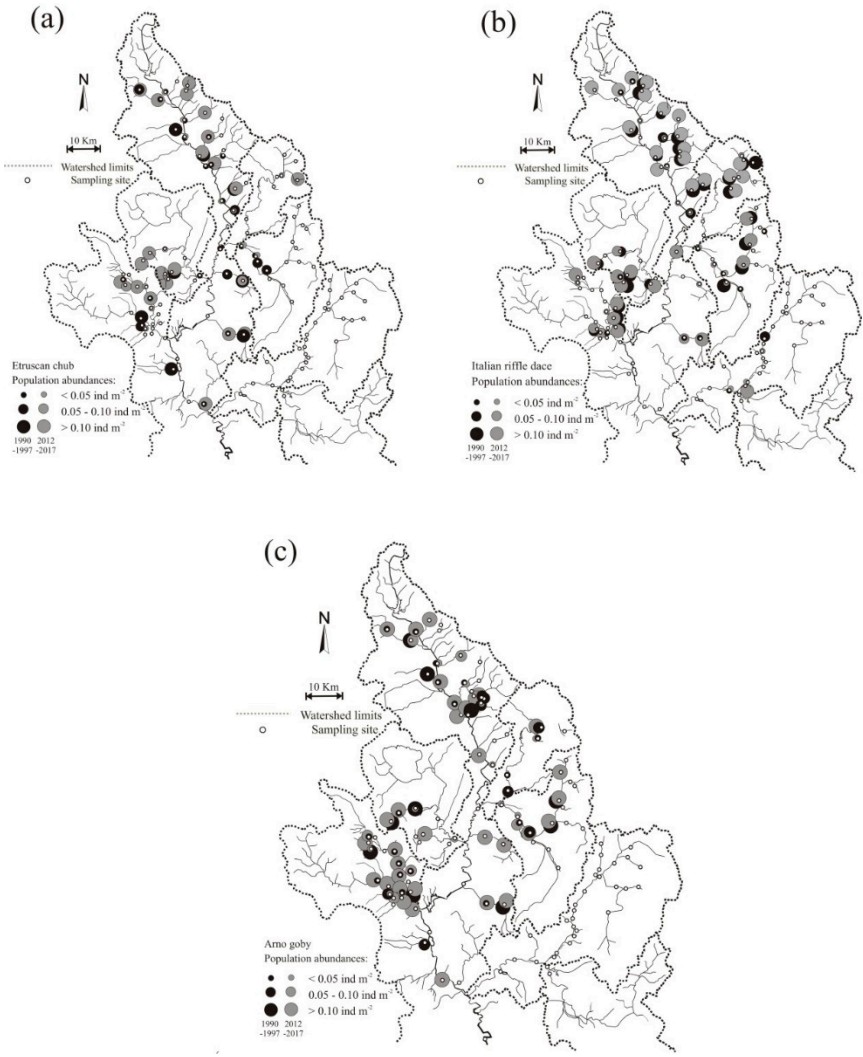

**Figure 4.** Comparison of the distribution and population abundances between the periods 1990–1997 (black circles) and 2012–2017 (grey circles) for: (**a**) *Squalius lucumonis*; (**b**) *Telestes muticellus*; (**c**) *Padogobius nigricans*.

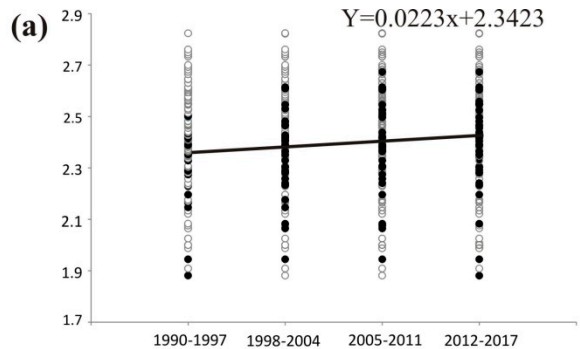

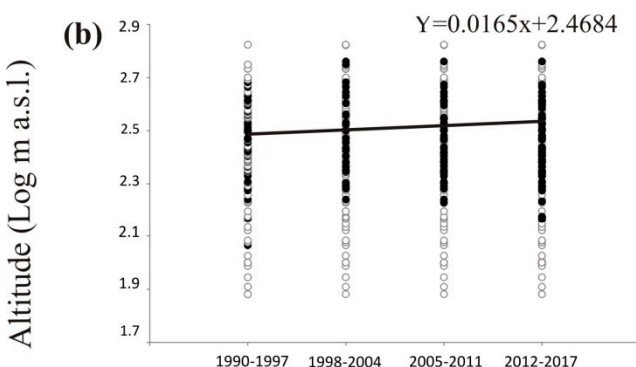

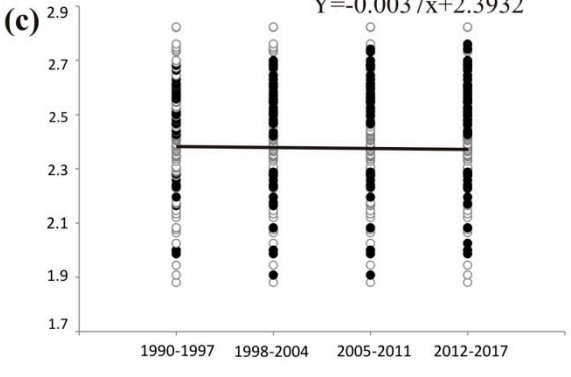

Sampling period (Years)

**Figure 5.** Occupancy-elevation plots showing occupied (filled circles) and unoccupied (clear circles) sites for each time period for: (**a**) *Squalius lucumonis*, (**b**) *Telestes muticellus* and (**c**) *Padogobius nigricans*. Solid lines represent the linear regressions.

The axes 1 and 2 of CCA explained 77.6% of the total variability and the analysis resulted highly statistically significant at the Monte Carlo test ($F = 6.75$, $p = 0.001$, total inertia = 2.996). Except for the pH, all environmental parameters resulted significantly related to the first axis of CCA, which represented the longitudinal gradient of the rivers (Table 1). Altitude, average current speed, distance from the source, flow rate, river fragmentation degree and watershed area resulted significantly inversely related to Axis 2, whilst chlorides, conductivity, $PPO_4$ and $SO_4$ were positively related to the same axis (Table 1). The species projection points showed for *S. lucumonis* and *T. muticellus* the strong link with elevation. The proximity of the points corresponding to *S. lucumonis*, *T. muticellus* and *P. nigricans* confirmed that the three species often occur together (Figure 7).

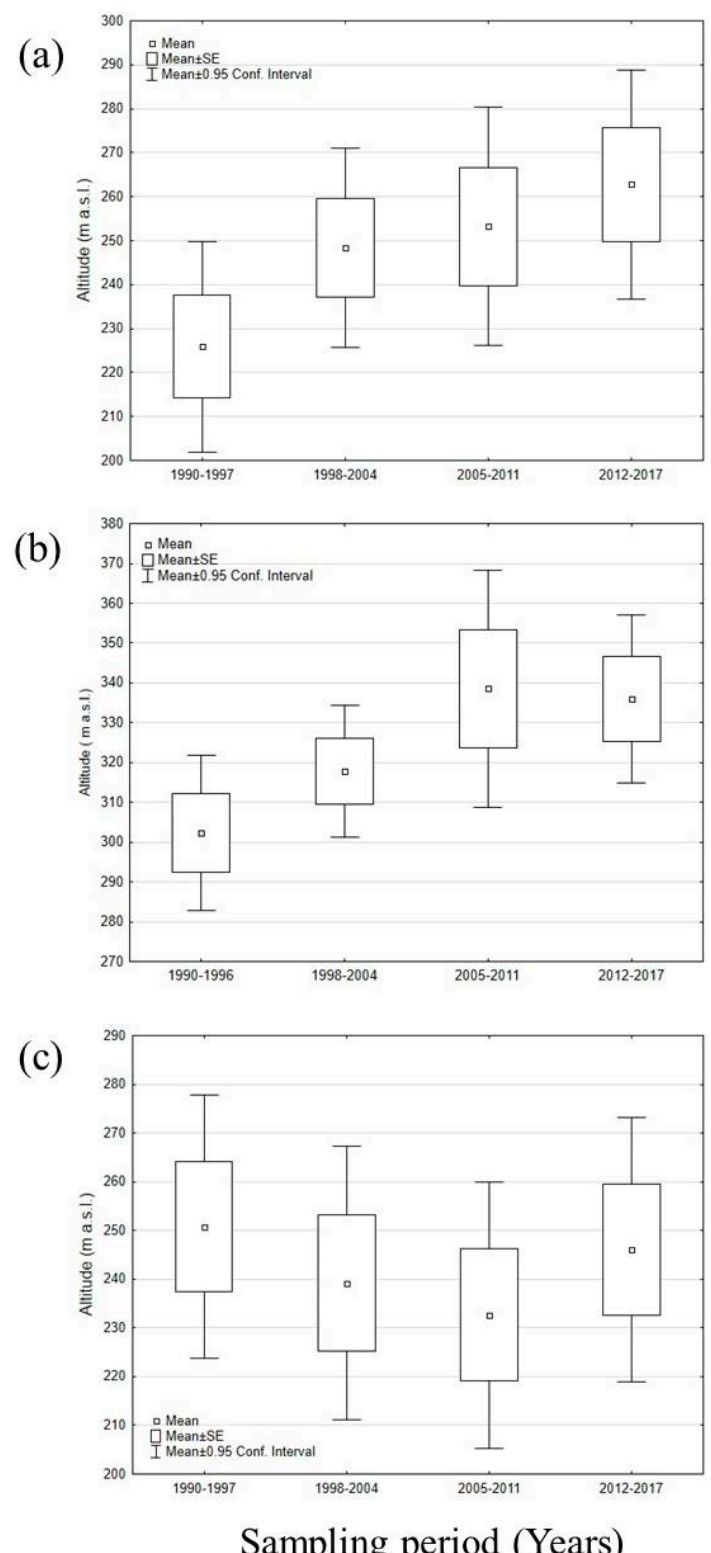

**Figure 6.** Trend over time of the mean altitude values for: (**a**) *Squalius lucumonis*, (**b**) *Telestes muticellus* and (**c**) *Padogobius nigricans* considering only the occurring sites in the census periods 1990–1997, 1998–2004, 2005–2011, 2012–2017.

**Table 1.** Canonical and correlation coefficients of environmental variables with axis. $p < 0.05$ is in bold.

| Environmental Parameters | AX1 | $p$ | AX2 | $p$ |
|---|---|---|---|---|
| Altitude (m a.s.l.) | **0.802** | **0.000** | **−0.225** | **0.000** |
| Average current speed (m s$^{-1}$) | **0.183** | **0.002** | **−0.336** | **0.000** |
| Cl (mg L$^{-1}$) | **−0.719** | **0.000** | **0.477** | **0.000** |
| Conductivity (S cm$^{-1}$) | **−0.738** | **0.000** | **0.399** | **0.000** |
| Dissolved oxygen (mg L$^{-1}$) | **0.193** | **0.001** | −0.004 | 0.943 |
| Distance from the source (km) | **−0.652** | **0.000** | **−0.544** | **0.000** |
| EBI (units) | **0.537** | **0.000** | −0.111 | 0.061 |
| EBI Quality Class (units) | **−0.606** | **0.000** | 0.111 | 0.062 |
| Flow rate (m$^3$ s$^{-1}$) | **−0.135** | **0.023** | **−0.619** | **0.000** |
| NNH$_3$ (mg L$^{-1}$) | **−0.317** | **0.000** | 0.021 | 0.728 |
| pH (units) | −0.013 | 0.826 | 0.115 | 0.052 |
| PPO$_4$ (mg L$^{-1}$) | **−0.259** | **0.000** | **0.122** | **0.040** |
| River fragmentation degree (units) | **0.156** | **0.008** | **−0.400** | **0.000** |
| SO$_4$ (mg L$^{-1}$) | **−0.593** | **0.000** | **0.127** | **0.033** |
| Water temperature (°C) | **−0.249** | **0.000** | −0.097 | 0.104 |
| Watershed area (km$^{-2}$) | **−0.547** | **0.000** | **−0.700** | **0.000** |

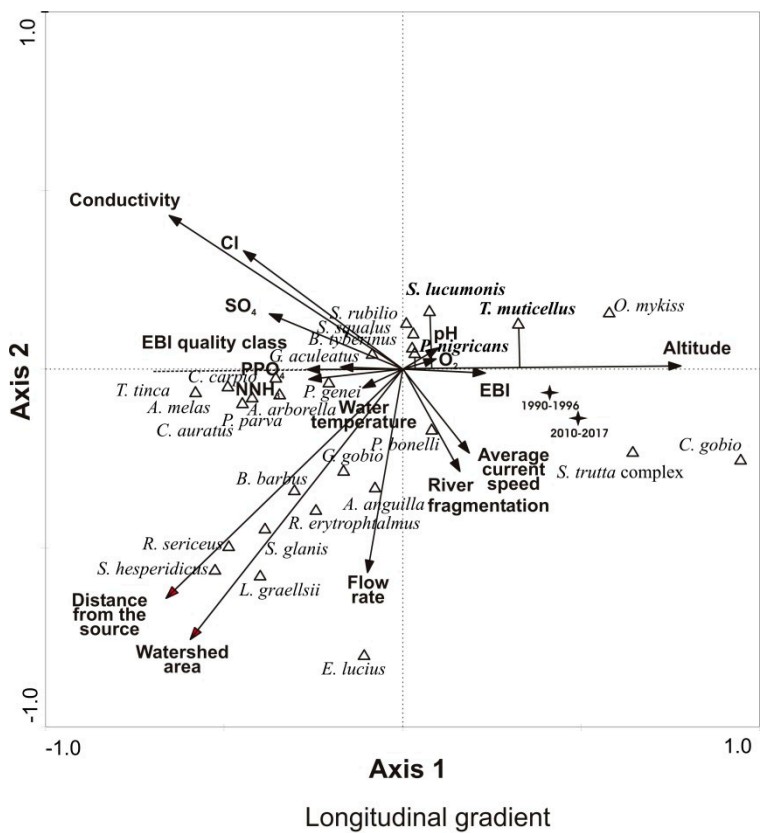

**Figure 7.** Canonical Correspondence Analysis results: biplot of fish and environmental variables for the census periods 1990–1997, 1998–2004, 2005–2011, 2012–2017. Black stars denote centroids of the periods 1990–1996 and 2010–2017, calculated as average scores of the samples belonging to those years.

For *S. lucumonis* and *T. muticellus* the GLM results showed a progressive shift of the species towards river stretches located at higher altitudes, confirming their tendency to move upstream along the longitudinal gradient of the river (Figure 8a,b). For *P. nigricans* the GLM results showed a progressive increase of population abundances over time, whilst they did not result a significant variation in altitude preferences of the species (Figure 8c).

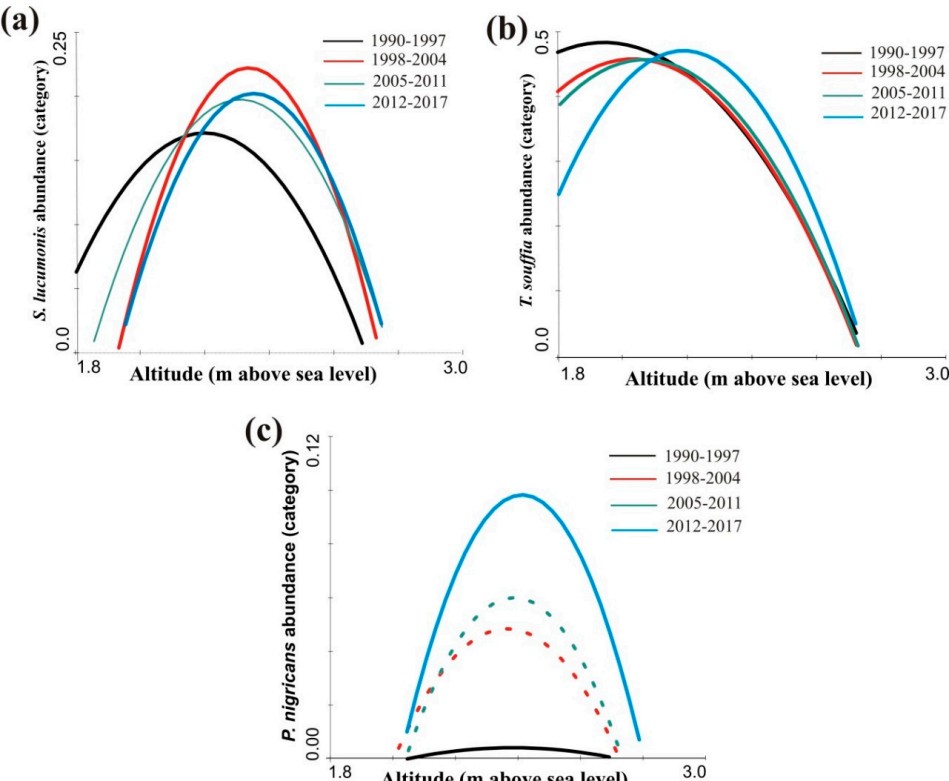

**Figure 8.** The effects of altitude on the abundance of (**a**) *Squalius lucumonis*, (**b**) *Telestes muticellus* and (**c**) *Padogobius nigricans* over the census periods 1990–1997, 1998–2004, 2005–2011, 2012–2017. Solid lines indicate significant results from the general linear model (GLM).

The distribution map of river barriers in the study area showed the presence of 188 weirs of more than 80 cm height (referred to the physical structure) and seven dams of more than 20 m height located in the whole hydrographic network (Figure 9).

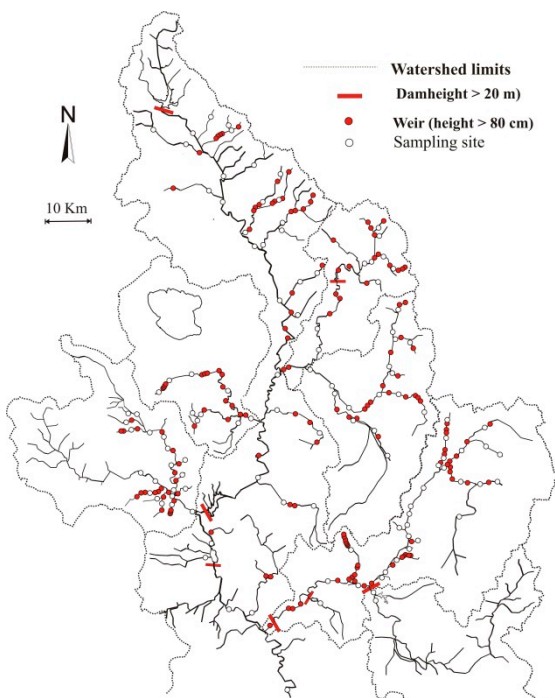

**Figure 9.** Weirs and dams' locations in the study area.

### 3.3. Population Status

Population density increased over time for all three species; except for *S. lucumonis* (Figure 10a), for the other species repeated measures ANOVA showed highly significant differences between periods ($p < 0.01$) (Figure 10b,c). As regards the body condition trend over time, the differences between periods were statistically significant at the repeated measures ANOVA only for *S. lucumonis* ($p = 0.005$), even if the mean values resulted in always being below the lower limit of the optimal range 95–105 (Figure 10a). The differences over time of mean PSD values resulted significant for *T. muticellus* ($p = 0.013$), with the average values calculated for the periods 2005–2010 and 2011–2017 falling within the optimal range of 35–65. In the case of *P. nigricans*, the results of repeated measures ANOVA revealed a significant, progressive decrease in average values over time ($p = 0.002$), with the PSD values consistently above the optimal range, indicating the lack of young specimens in the sample (Figure 10c).

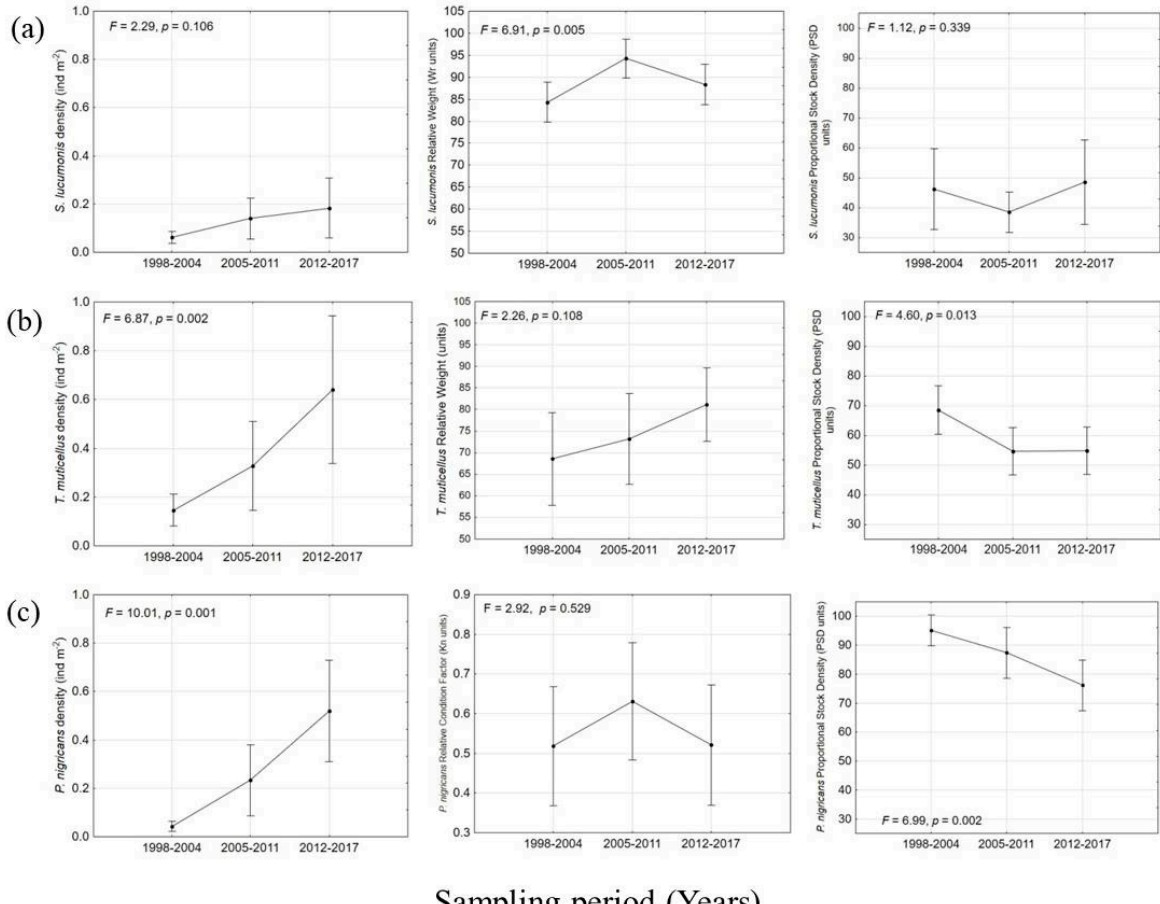

**Figure 10.** Trend over time of the mean density, relative weight (Wr), and proportional stock density (PSD) values for: (**a**) *Squalius lucumonis*, (**b**) *Telestes muticellus* and (**c**) *Padogobius nigricans*, for the sampling periods 1998–2004, 2005–2011, 2012–2017. Vertical bars denote 0.95 confidence intervals. Note the different Kn scale values compared to Wr.

### 3.4. Occupancy, Colonization, Extirpation and Detection Probabilities

Detection probability was high for all three species ranging from 0.79 (*P. nigricans*) to 0.74 (*S. lucumonis*), with no results of perfect detection (1.00). Analyzing the detectability values for the 4 time periods, some differences emerged; in particular, the values of detectability were lower during the first period for all species, even if the values never fall below 0.5 (Table 2). The occupancy probabilities ranged from 0.36 for *S. lucumonis* to 0.49 for *T. muticellus*. The probability of colonization are resumed for all species considered equal to about twice the probability of extinction, which fell within the range

0.10–0.13 (Table 3). Elevation and river fragmentation were key predictors of occupancy, colonization, extirpation and detection, as they were included in the best multi-season models, within two AIC units of the top model (Table S1). For *S. lucumonis*, estimated extinction probabilities decreased from low to high-elevation sites, while the occupancy probability progressively and linearly decreased from low to high-fragmentation degree sites (Figure 11a). For *T. muticellus*, the extinction probability decreased with elevation and progressively increased with the river fragmentation degree; the occupancy probability increased with elevation (Figure 11b). For *P. nigricans*, the colonization probability decreased with elevation (as well as occupancy probability) and river fragmentation degree (Figure 11c).

**Table 2.** Overall occupancy (ψ), colonization (γ), extirpation (ε) and detection (p) probabilities (±SE) for *Squalius lucumonis*, *Telestes muticellus* and *Padogobius nigricans*.

| Species | Occupancy (ψ) | Colonization (γ) | Extirpation (ε) | Detection (p) |
|---|---|---|---|---|
| *S. Lucumonis* | 0.36 ± 0.06 | 0.22 ± 0.06 | 0.11 ± 0.06 | 0.74 ± 0.04 |
| *T. Muticellus* | 0.49 ± 0.06 | 0.28 ± 0.08 | 0.13 ± 0.05 | 0.75 ± 0.03 |
| *P. Nigricans* | 0.40 ± 0.06 | 0.21 ± 0.07 | 0.10 ± 0.03 | 0.79 ± 0.03 |

**Table 3.** Probability of detection (± standard error (SE)) across the four time periods for *Squalius lucumonis*, *Telestes muticellus* and *Padogobius nigricans*.

| | Detection Probability | | | |
|---|---|---|---|---|
| Species | Time Period 1 | Time Period 2 | Time Period 3 | Time Period 4 |
| | 1990–1997 | 1998–2004 | 2005–2010 | 2011–2017 |
| *S. Lucumonis* | 0.57 ± 0.01 | 0.82 ± 0.05 | 0.76 ± 0.06 | 0.65 ± 0.07 |
| *T. Muticellus* | 0.64 ± 0.01 | 0.75 ± 0.05 | 0.69 ± 0.06 | 0.81 ± 0.05 |
| *P. Nigricans* | 0.61 ± 0.01 | 0.73 ± 0.06 | 0.78 ± 0.05 | 0.86 ± 0.05 |

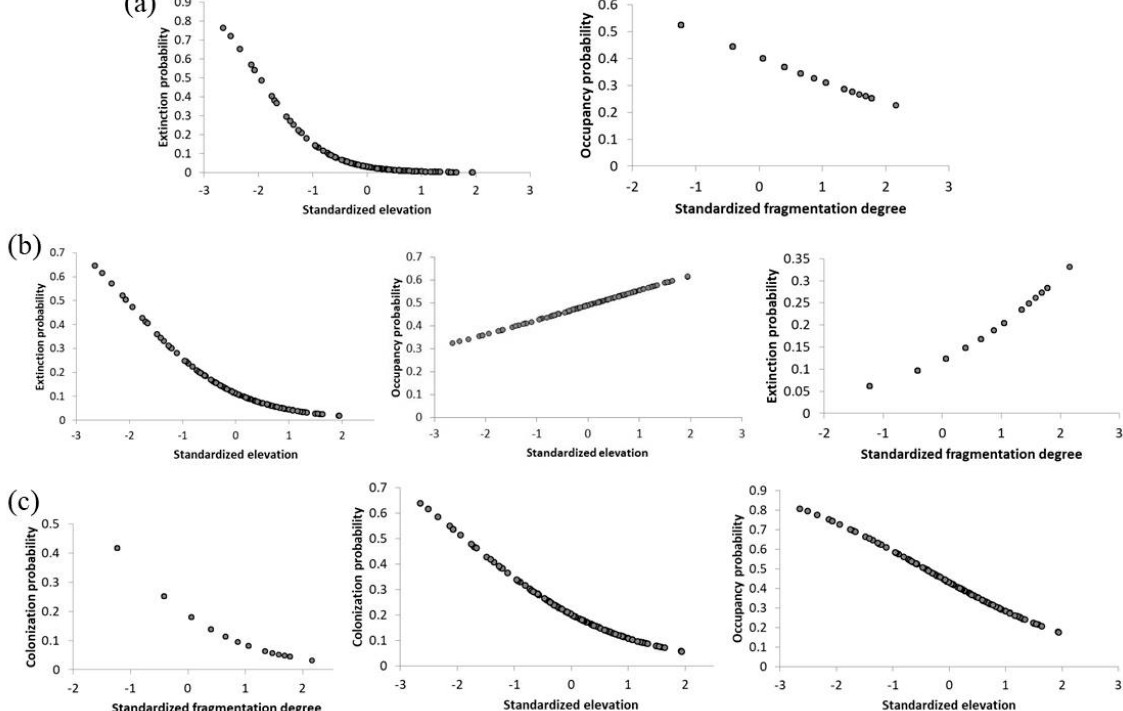

**Figure 11.** Effects of elevation and degree of river fragmentation on occupancy, colonization and extinction probabilities from the best informative models for: (**a**) *Squalius lucumonis*, (**b**) *Telestes muticellus* and (**c**) *Padogobius nigricans*.

## 4. Discussion

The increasing water temperature over time and the clear decreasing trend of annual average flow rates provided some evidence that climate changes have induced direct effects on the watercourses in the study area, and consequently affected the freshwater fish inhabiting these environments. In particular, the increases in water temperature could be reasonably related to the spatial distribution shifts of the endemic *S. lucumonis* and *T. muticellus*, which tend to move further upstream in order to reach their thermal optimum [27,61]. Over the last 27 years, within a proven period of climate warming and decreasing flow rates, both species have shown significant range shifts to higher elevation resulting in a mean increase of 40.5 and 33.7 m for *S. lucumonis* and *T. muticellus*, respectively. These findings are consistent with the mean altitudinal range shifts (32.7 m) reported for 47 freshwater fish species in a previous study carried out on British inland waters under climate warming conditions [21]. By contrast, the absence of significant variations in water temperature in their occurring sites supported the hypothesis that the distribution shift along the longitudinal gradient of rivers allowed *S. lucumonis* and *T. muticellus* to remain within their thermal preferences; tracking thermal preferences has also been found in marine fish species, as reported by Nye et al. [62] for the Northeast USA fish stocks. Beside this, at the moment there is no evidence for a reduction in the distribution range for either species but on the contrary there is a slight increase in the number of sites where the species are present, compared to the first recording period. This result suggests that, at this stage, climate changes have not yet led to decreased population density and local extinction phenomena of these species adapted to the Mediterranean climate. Moving upstream, however, species are exposed to a greater risk of extinction, since the Apennines watercourses of modest dimensions could dry out in their upper stretches during drought seasons in particular [9].

Amongst climate-induced effects, the combination of drought, loss of habitat and river fragmentation can be considered strong threats for inland waters biodiversity [11,63]. Due to the considerable fragmentation of the hydrographic network in the Tiber river basin, the loss of flow occurring in drought periods can lead to the isolation of species with limited thermal tolerance who would move in search of more suitable habitat conditions. In this context, species characterized by poor vagility are more susceptible to extinction in comparison with those of greater dispersal ability [64]. This is the case of *P. nigricans*, a sedentary bottom-dwelling species characterized by limited vagility [65–67]. The species has not shown the ability to perform significant movements, probably also due to the numerous insurmountable weirs and dams located on the watercourses in the study area, which are hindering the species' migration along the longitudinal gradient of the rivers. The future status of *P. nigricans* populations is of particular concern, as in the Tiber river basin the species is likely to remain trapped in environments where it must face the increase in temperature and other multiple anthropogenic stressors such as water pollution and the presence of the invasive *P. bonelli*. The significant increase in water temperature observed over time in the sites in which the species was present confirmed this hypothesis. However, except for the local extinction events that occurred as a result of unprecedented drought periods, which concern some sites within the Nestore River basin, for *P. nigricans* an overall increase over time of the total number of occurring sites also was observed and, as shown by the GLM analysis, a progressive increase in population density. These surprising results supported the hypothesis that in the short term, in some cases the fish species could benefit temporarily from increasing temperatures in terms of increased recruitment and did not show apparent changes in their distribution [7,62]. Certainly further research are needed to identify the thermal maxima tolerable by the species and to predict when it will be reached.

The multi-season models results suggested that *S. lucumonis* and *T. muticellus* extinction probabilities were highest at low-elevation sites, while for *P. nigricans* the major obstacle to the colonization of other sites seems to be represented by the river fragmentation, as well as by the altitude. As already reported in previous studies concerning salmonid fishes [68], also in our research the altitude seems to play a key role in the distribution of the species inhabiting the Tiber River basin, also in consideration of the close link between altitude, water temperature and presence of alien species.

However, unlike the cold-water species, for which recent studies demonstrated significantly greater probability of extinction than the probability of colonization, and showed marked reductions in their range [68,69], in this case the opposite happens. This result seems to confirm the hypothesis that for the cool-water species the effects of climate change are starting to occur but it will take some time before the striking consequences could be highlighted.

Similarly to previous studies [28,45], our CCA demonstrates how the fish species are distributed along the longitudinal gradient of the watercourses and which are the environmental variables that most influence their distribution. As reported for other watersheds in the Mediterranean area [70], also in the Tiber River basin the longitudinal gradient was associated with a progressive decline in water quality, proceeding from upstream to downstream, demonstrated by the decrease of EBI and increased conductivity and concentration of dissolved salts. Beside this, recent studies carried out in the Tiber river basin [28,71] showed the presence of many invasive species, mainly located in lowland stream reaches and lakes [72,73]; these species are often limnophilous and can be favored by climate changes due to their wider ecological tolerance.

Due to the presence of the aforementioned multiple anthropogenic stressors, the results of studies on climate change effects should be interpreted with caution, because there are many variables to be considered in order to evaluate biological changes [64,74,75]. Moreover, synergistic effects that lead to an amplification of negative impacts on freshwater fishes can often occur [76]. These negative impacts can result in fish population-level changes [77]. *S. lucumonis* and *T. muticellus* resulted in good population status in terms of abundance and age structure, but not in terms of relative weight; probably the poor body conditions not made them able to adapt to the new environmental conditions and could represent the reason that pushes them to search for more suitable sites, adapting their distributions. For *P. nigricans* the unbalanced population structure, with the lack of young specimens, could represent a first sign of the state of uneasiness of the species.

## 5. Conclusions

The present research has shown fish distributional changes in a context of climate warming; these results, based on observational data, can be useful for planning future management scenarios [78], also in light of the fact that the climate models predict further strong impacts on freshwater fish [64]. In particular, the possible conservation strategies that can be suggested for fish biodiversity conservation under climate-induced impacts are: (i) the maintenance of ecological flow, whereas the natural variability of the flow rate is indispensable for the conservation of habitat suitable for the life of freshwater fishes [3]; (ii) the restoration of river connectivity in order to provide opportunity, to fish species with low thermal tolerance, to move towards thermal optimum considering, however, the possibility of maintaining barriers in the event of a need to prevent the rise of exotic species; (iii) the establishment of protection areas where total autochthonous assemblages are preserved and thus play a key role in maintaining biodiversity; (iv) improvement of water quality in polluted rivers is needed to reduce additional anthropogenic stressors for resilient fish populations to adapt to climate change; (v) the restoration of riparian areas to provide shade and have directly cooling effect on river temperature [79].

**Supplementary Materials:** The following are available online at http://www.mdpi.com/2073-4441/11/11/2349/s1, Table S1: Models within two Akaike Information Criterion (AIC) units of the top model for estimating occupancy (ψ), colonization (g), extirpation (ε) and detection (p) probabilities for (a) *Squalius lucumonis*, (b) *Telestes muticellus*, (c) *Padogobius nigricans*.

**Author Contributions:** All authors contributed substantially to conceptualization, methodology, statistical analysis, investigation and interpretation of data. All authors give approval of the final version of the manuscript.

**Funding:** This research received no external funding.

**Acknowledgments:** The authors would like to thanks all the people who participated in the field activities to collect data for the Regional Fish Map of the Umbria Region and all the students and collaborators who joined the project. The authors are grateful to the reviewers for their valuable comments that considerably improved the original version of the manuscript.

**Conflicts of Interest:** The authors declare no conflict of interest.

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
