# Peer review of "Endemic Freshwater Fish Range Shifts Related to Global Climate Changes: A Long-Term Study Provides Some Observational Evidence for the Mediterranean Area"

_water, doi:10.3390/w11112349_

Round 1
Reviewer 1 Report
Overall this research is timely and understanding how freshwater fish respond to climate change is important. This is an extensive dataset with good temporal and geographical scope. There are some downfalls that could have been addressed with a better network of temperature loggers. The results are comprehensive with some nice maps and figures presenting the data. Some figures need editing to be made more legible, with specific edits attached the the document. The discussion nicely frames the results, there are a few sections where a negative suggestion has been made, however, most of these results show that the fish are adapting and that the populations aren't declining so the framing needs to be done carefully. Additionally it would be interesting to see a discussion of the current knowledge gaps in the introduction and discussion on how fish move around the barriers and their thermal envelopes.

Author Response
Response to Reviewer 1 Comments
Point 1: line 15 – This abstract starts with the aims, it would be helpful for the reader to have a short introduction to this topic and why it is important https://cbs.umn.edu/sites/cbs.umn.edu/files/public/downloads/Annotated_Nature_abstract.pdf
Response 1: a short introduction to the topic was added to the abstract (lines 15-18 of the revised MS).
Point 2: line 22 Avoid starting a sentence with the genus abbreviation, so either list Padogobius in full in this instance or reorder the sentence.
Response 2: starting a sentence with the genus abbreviation was avoided throughout the text
Point 3: line 25 – local extinction = extirpation
Response 3: “local extinction” was replaced by “extirpation” throughout the text.
Point 4: line 30 consider using “climate change” instead of “global warming”, as climate change includes the warming or air and water but also takes into account changing patterns of precipitation which may lead to drought. As your abstract includes drought and water temperature, I would recommend climate change to be the most useful terminology. Also the manuscript only refers to global warming twice but to climate change 6 times. Consider revising the two uses of global warming for consistency.
Response 4: “climate change” was used instead of “global warming” throughout the text.
Point 5: line 39 Could remove this sentence and instead edit the following sentence to;“Habitat alteration, invasive species, water pollution and abstraction and overfishing are amongst the main threats [3] leading to 56% of endemic freshwater fish in the Mediterranean basin being threatened [2].”
Response 5: the sentence was removed and edited as suggested by the referee (lines 42-44 of the revised MS).
Point 6: line 53 – Surely water availability is also key, e.g. some rivers drying up, or new water bodies becoming available due to snow melt. http://www.publish.csiro.au/MF/MF10303 https://www.nature.com/articles/nature04141
Response 6: water availability and two references were added to the climate-induced effects (line 55).
Point 7: line 68 citation needed – possibly; https://www.feem.it/en/publications/feem-working-papers-note-dilavoro-series/the-water-abstraction-license-regime-in-italy-a-case-for-reform/ or https://www.eea.europa.eu/publications/92-9167-205-X/page013.html showing increased area of irrigation between 1985 to 1996
Response 7: as suggested by the referee, two citations on increased water abstraction were added (line 70).
Point 8: line 79 Reference needed for habitat preferences; 84 – 86 Reference needed for water pollution and high exotics in lower reaches
Response 8: Three references for habitat preferences, water pollution and many exotic species in downstream reaches were added (lines 82 and 92).
Point 9: Introduction missing actual summary of what thermal niches each of the three species can actually inhabit, or a mention that this is unknown. Apart from uncited sentence on riffle dace liking “cool waters”
Response 9: To specify that no detailed information is currently available on the thermal niche of the three species, but many studies showed their location along the longitudinal gradient of the rivers, the following sentences and citations were added: “Currently there are scarce information in the literature on the thermal niche of the three species; however many studies showed their reophilic characteristics and their location in the middle-upper section of the rivers, where current speed and dissolved oxygen are quite high and water is mildly cool [27-29].” (Lines 89-92).
Point 10: One of the flaws of this data collection is that there is no collection of temperature data along an elevation or longitudinal gradient, so dispite the results suggesting the moving to upstream, or to higher elevation is due to climate change, temperature, there is no way to actually link the two, this could have been achieved with some cost-effective HOBO temperature loggers in each river. Temperature wasn’t even recorded for every basin.
Response 10: Unfortunately water temperature data are not available along the longitudinal gradient for all the rivers. However, previous studies carried out in the study area provided evidence of a strong link between elevation and thermal conditions, then in the present research we considered the altitude as a good indicator of thermal conditions in the Tiber river basin.
Point 11: line 137 give the manufacturer for the flow measurer, as given for YSI
line 137 – 138 – presumable water samples were taken to then measure these in the lab, what volumes of water were measured, how many samples, were they taken from the water surface or at depth? Was the depth consistent? I understand the desire to not replicate a method here, however, as the reference publication is in Italian and the readership of this is international I would highly recommend providing a summary of the methods here.
Response 11: the method for the collection of water samples was specified. Two references in English languages were added for the laboratory analysis. The mean depth of the river stretches was also given (lines 142-147).
Point 12: line 139 what was the source of these digital maps?
line 143 – so these were visited in the field? What was the accuracy of the GPS and the model? lines 144 – 145 How was the height of weirs measured? And what was measured, the physical structure or from water height?
Response 12: clarifications regarding the detection of weirs have been added (lines 148-153 and 378-384).
Point 13: line 186 – 189 analysis done with what software?
lines 190 – 194 again what software and version is used?
line 217 GLM, and ANOVA etc done in what software?
Response 13: the software used for data processing has been specified (lines 203, 224, 234-236.
Point 14: lines 224 – 226 I wouldn’t refer to the sampling years as seasons, that could lead to confusion over seasons autumn, winter, spring, summer. Could you use the word period instead?
Response 14: The word “period” was used instead of “season” throughout the text.
Point 15: Figure 2 – Would it be more logical to list Clitunna River 1 at top going down to Tiber River 3 at the bottom? I would suggest making all x-axis with comparable dates. X-axis title needs to read “MonthYear” and can just be on the bottom figure. Y-axis should all be formatted the same from 0 – 35, I think the 5 increment looks the best. Consider moving information from the legends on y, r, p values etc. either into the text or into the figure legend as they are hard to read in the graphs themselves. Also if using labels a, b and c then remove the labels above each graph of ‘Clitunno River 1’ etc. The figure legend contradicts the results text here states Topino and Tiber data starts 2004, in text line 252 it says 2005.
Figure 3 – legend refers to a – e but these are not indicated in the figure. As title for y axis is the same across all graphs consider removing repetition and just have one large rotated title to make it clear. Same for Year. Consider making the y scales the same so the graphs are comparable, or at least make “a” and “b” the same and c – e the same. Explain in the figure legend what the solid and dashed lines represent. Also keep consistent the tab lines on the x-asix i.e. for c) they are every 10 years but on e) they are every 20 years.
Figure 4 – Legend contradicts text on Line 266 1997 and in legend 1996. Legend says 2012 – 2017 but in figures it shows 2008 – 2017 – please clarify. Also dates missing from b). Also figure states grey and black show occupied vs unoccupied but in the figure the colours are below the different time periods. There is a huge amount of information in this figure and the graphs with Altitude and Sampling period are hard to see and compare because they are small, I think this data would be better displayed in a separate figure with the three graphs next to each other. For example cannot see the axis values for occupancy-elevation plots – are they the same scale or not. Also would it be suitable to add a trendline to the occupancy-elevation plots? I’m not sure what it means when there is a sampling site point but no data, is the smallest circle size actually anything greater than zero to 0.05, therefore white sample site dots are where zero of that species were found – this needs to be made clear to the reader.
Response 15: Figures 2, 3 and 4 have been revised according to the suggestions of the referee. A new Figure 5 was provided.
Point 16: lines 285 – 289 Would the decrease be more comparable between Tiber and Chiani and Topino if the same time frame were used? Data from Tiber 1 and Tiber 2 look very similar, could one of these plots instead have a comparable time frame to Ciani and Topino instead?
Response 16: we tried to use comparable time periods (1992-2016) for Tiber1, Chiani and Topino rivers but the regression resulted not significant. Furthermore, data relating to some years ('92 -'94) are missing for Tiber 1 and 2.
Point 17: lines 290 – 292 Along with reporting that the ANOVA was significant the test value (F) should be reported and the degrees of freedom see example in https://link.springer.com/article/10.1007/s10750-018-3644-6
Response 17: ANOVA results were reported according to the suggestion of the referee (lines 313-315).
Point 18: lines 297 – 315 – It’s not clear to me why the temperature across periods is reported for each species of fish. Can this be more explicit in the wording.
Response 18: The temperature across periods is reported for each species in order to test the hypothesis that the thermal conditions of the sites inhabited by the species reflects their ability to shift. Then, our results showed that water temperature varied for P. nigricans since the species is not able to move, while remains constant for T. muticellus and S. lucumonis, probably because they are able to move to reach their thermal optimum. This has been made explicit in the text (lines 319-325).
Point 19: Figure 6 Use full wording for CCA in the legend, not the abbreviation. Bioplot of fish presence?
Figure 7 – Legend could be reworded to have a better structure. e.g. The effects of altitude no the abundance of a) Squalius lucumonis…. over the census periods… Solid lines indicate significant results from the General Linear Model. Description for the acronym M a.s.l. needed in the figure legend. Y axis in brackets says units, what are the units.
Figure 9 – inconsistency of reported number of decimal places for F values, sometimes there are three sometimes only 2, these should be consistent, I would suggest reporting 2 decimal places. Legend needs to explain all axis abbreviations such as Kn, PDS and indicate units. Repeated axis labels can be removed – Samping period (years) can just be on C row for example. Why is 95% confidence interval the error bar of choice here but earlier plots Figure 5 showing mean + SE and 1.96 SE? My preference is 95% CI, consider using this in Fig 5 too. Consider making y-axes comparable between species, other wise it looks like there has been a large increase in mean density of S. lucumonis but actually it just has a smaller range of densities displayed. Where that may not be possible e.g. the Kn for P. nigricans as it may make the data not easible visible as the current plot is 100 times different scale, make a note in the legend that the scale is varied so the reader is aware. Legend states from 1997 - 2004 but the plots show 1998 – 2004.
Response 19: Figures 7, 8 and 10 have been revised according to the suggestions of the referee.
Point 20: lines 399 – 402 This sentence need rewording, surely a detection probability of 1 is actually extremely unlikely. Suggested revision; “Detection probability was high for all three species ranging from 0.79 (P. nigricans) to 0.74 (S. lucumonis), with no results of perfect detection (1.00).
Response 20: The sentence was revised according to the suggestion of the referee (lines 400-402).
Point 21: lines 408 – 408 This is key information and needs to be made clear for the reader. Revise as appropriate; “Elevation and river fragmentation were key predictors of occupancy, colonisation, extirpation and detection”.
Response 21: The sentence was revised according to the suggestion of the referee (lines 406-407).
Point 22: Table 2 – Legend needs to state what error is being reported here – standard error? 95% confidence? Also species names should be given in full in either the legend or the table.
Table 3 – Same comment as earlier on use of term season. Species name in full in either legend or table. What is the error range reported?
Response 22: Table 2 and 3 were revised according to the suggestion of the referee.
Point 23: Overall I feel the results is missing a key link between what temperatures the fish were actually found in to show the thermal envelope each year. The only nod to this data is the CCA plot but water temperature is only significant for one axis and the directional arrow is very short.
Response 23: As mentioned above, based on previous studies, in the absence of water temperature data along the upstream – downstream gradient of the rivers, we considered the altitude as a strong indicator of thermal conditions, in the study area.
Point 24: Table 4 – as there are already a large number of tables and figures in this manuscript could this be in supplementary information instead of in the main document? Is it necessary to report 4 decimal places for AIC weight and Model likelihood? If you do need 4, then be consistent as in some instance you have 3 decimal places reported. Report species names in full.
Response 24: Table 4 was revised according to the suggestion of the referee and it was moved to the supplementary material.
Point 25: Figure 10. 4 decimal places not needed on the x axis for Standardised elevation.
Response 25: Figure 11 has been revised according to the suggestions of the referee.
Point 26: line 420 I would say it is not clear that it is climate change that has driven a negative change, also what is being referred to here as negative is not clear. I’m not sure that it’s clear to say range shift it ‘negative’, if the fish are able to adapt, surely that is ‘positive’. Can this term be changed for a less subjective phrase? The rest of the paragraph goes onto state there hasn’t been range contraction, decreasing densities or extinction, therefore the initial summary of ‘negative’ doesn’t seem to be summarised here.
Response 26: the word “negative” has been deleted.
Point 27: lines 422 – 423 Need a reference for the statement ‘tend to move further upstream in order to reach their thermal optimum’ have you even proved here that it is cooler upstream than it is downstream?
Response 27: two references were added (line 429).
Point 28: line 428 I suspect the wrong citation has been given here as reference 49 is a modelling book and from the contents I can’t see anything about fish range shifts in there. This needs checking
Response 28: The correct reference was provided.
Point 29: line 429 No tables, figures or text adequately show the results that is discussed here that ‘temperature in their occurring sites supported the hypothesis that the distribution shift… allowed them to remain in their thermal preferences’ and equally the thermal preferences haven’t been stated.
Response 29: The sentence refers to the results reported in lines 325-331, where the thermal ranges for each species have been stated.
Point 30: line 431 – ‘as reported by Nye et al.’ in this instance it reads as if this author also studied the longitudinal gradient of rivers and these two species, when in fact this was a study of marine fish so not directly comparable. This needs rephrasing to emphasise this is a different system and different species i.e. “tracking thermal preferences has also been found in marine fish species [51].” Or is there a citation on rivers that would be more comparable?
Response 30: the sentence was rephrased according to the suggestion of the referee (line 438).
Point 30: lines 436 – 438 why does moving upstream expose greater risk to extinction – citation needed.
Response 30: A citation was provided to support this sentence (line 445).
Point 31: lines 445 – 459 so despite predictions of its low vagility making it vulnerable to isolation and climate change it has increased it’s density and number of occupant sites. Seems to me the missing knowledge is what is it’s thermal maxima that it can survive? I think a discussion here of further research needed on thermal maxima and predicting when the temperature will reach those, is there a tipping point for example.
Response 31: the following sentence on further research needed “Certainly further research are needed to identify the thermal maxima tolerable by the species and to predict when it will be reached.” was added (lines 466-467).
Point 32: lines 486-489 this sentence needs rephrasing, suggests they couldn’t adapt their distributions, but your results suggest the opposite.
Response 32: the sentence was rephrased (lines 494-496).
Point 33: line 501 “improvement of water quality in polluted rivers” doesn’t seem like a very strong ending statement, consider revising. Suggested “improvement of water quality in polluted rivers is needed to reduce additional anthropogenic stressors for resilient fish populations to adapt to climate change.”
Response 33: the sentence was revised according to the suggestion of the referee (lines 510-511).
Point 34: I feel the discussion lacks reference to what further research is needed, and no discussion of managing river temperatures or mitigating climate change – several publications on how riparian cover, and adjacent land shaded by trees can have directly cooling effect.
Response 34: a sentence on the advisability to mitigate river temperature by restoration of riparian areas was added (lines 511-512).
Point 35: line 573 – Pompei et al 2018 also seems relevant citation that is missing http://www.aquaticinvasions.net/2018/AI_2018_Pompei_etal.pdf
Response 35: the citation was added (lines 664-665).
Reviewer 2 Report
The article is very interesting and properly written. The authors address a very important issue of the impact of global warming (causing deterioration in ecological conditions and water quality) on the biodiversity of three freshwater fish species: Squalius lucumonis, Telestes muticellus and Padogobius nigricans from the Tiber River basin (Italy). Multidimensional analysis using fish and environmental data collected at 117 sites between 1990 and 2017 showed that various species can respond to climate change depending on their dispersal abilities. S. lucumonis and T. muticellus changed their location (they swam upstream) and P. nigricans did not move significantly upstream due to is limited ability to migrate.
My only comments on this work are minor editorial remarks:
Figure 4 - small charts in the upper right corner are completely illegible. The use of colors in legends and drawings would make them clearer. Figure 5: Please change the graph - there are different font sizes in the description. It is not necessary to give additional names of the species of fish appearing on each drawing. Please standardize the font size in all Figures. Please improve the References section according to the requirements of the Water magazine!In the case of journal names, abbreviations should be provided. Some items lack DOI - e.g. positions 66, 52. Please do not use “&” when listing the names of authors (item 62).
Author Response
Response to Reviewer 2 Comments
Point 1: Figure 4 - small charts in the upper right corner are completely illegible. The use of colors in legends and drawings would make them clearer.
Response 1: the occupancy-elevation plots were displayed in a separate Figure 5.
Point 2: Figure 5 Please change the graph - there are different font sizes in the description. It is not necessary to give additional names of the species of fish appearing on each drawing.
Response 2: Figure 6 (ex figure 5) was revised according to the suggestions of the referee.
Point 3: Please standardize the font size in all Figures.
Response 3: The font size was standardized in all Figures.
Point 4: Please improve the References section according to the requirements of the Water magazine! In the case of journal names, abbreviations should be provided. Some items lack DOI - e.g. positions 66, 52. Please do not use “&” when listing the names of authors (item 62).
Response 4: the References section was revised according to the editorial rules of the Water journal.